# Top-k Training of GANs: Improving GAN Performance by Throwing Away Bad Samples

**Samarth Sinha** *
University of Toronto
samarth.sinha@mail.utoronto.ca

**Zhengli Zhao** *
University of California, Irvine
zhengliz@uci.edu

**Anirudh Goyal**
Mila, Université de Montréal

**Colin Raffel**
Google Brain

**Augustus Odena**
Google Brain

## Abstract

We introduce a simple (one line of code) modification to the Generative Adversarial Network (GAN) training algorithm that materially improves results with no increase in computational cost: When updating the generator parameters, we simply zero out the gradient contributions from the elements of the batch that the critic scores as 'least realistic'. Through experiments on many different GAN variants, we show that this 'top-k update' procedure is a generally applicable improvement. In order to understand the nature of the improvement, we conduct extensive analysis on a simple mixture-of-Gaussians dataset and discover several interesting phenomena. Among these is that, when gradient updates are computed using the worst-scoring batch elements, samples can actually be pushed further away from their nearest mode. We also apply our method to recent GAN variants and improve state-of-the-art FID for conditional generation from 9.21 to 8.57 on CIFAR-10.

## 1 Introduction

Generative Adversarial Networks (GANs) [15] have been successfully used for image synthesis [32, 47, 4], audio synthesis [10, 11], domain adaptation [53, 48], and other applications [45, 27, 50]. It is well known that GANs are difficult to train, and much research focuses on ways to modify the training procedure to reduce this difficulty. Since the generator parameters are updated by performing gradient descent through the critic, much of this work focuses on modifying the critic in some way [1, 30, 33, 16] so that the gradients the generator gets will be more 'useful'. What 'usefulness' means is generally somewhat ill-defined, but we can define it implicitly and say that useful gradients are those which result in the generator learning a better model of the target distribution.

Recent work by [44] suggests that gradients can be more useful when computed on samples closer to the data-manifold – that is, if we tend to update the generator and critic weights using samples that are more realistic, the generator will tend to output more realistic samples. [44] achieves state-of-the-art results on the ImageNet conditional image synthesis task by generating samples from the generator, computing the gradient of the critic with respect to the sampled prior that generated those samples, updating that sampled prior in the direction of that gradient, and then finally updating the generator parameters using this new draw from the prior. In short, they update the generator and critic parameters using a $z'$ such that the critic thinks $G(z')$ is 'more realistic' than $G(z)$. However, this procedure is complicated and computationally expensive: it requires twice as many operations per gradient update. In this work, we demonstrate that similar improvements can be achieved with a much simpler technique: we propose to simply zero out the gradient contributions from the elements of the batch that the critic scores as 'least realistic'.

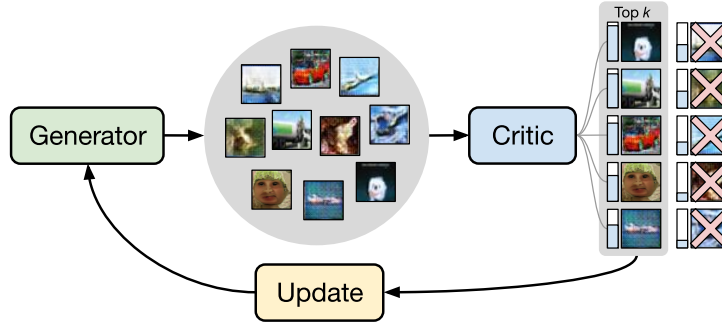

Figure 1: Diagram of top-$k$ training of a GAN. The generator generates a batch of samples, which are scored by the critic. Only the $k$ samples with the highest scores are used to update the generator.

Why should this help? In an idealized GAN, the trained critic would slowly lose its ability to tell which inputs were samples from the generator and which inputs were elements of the target distribution, but in practice this doesn't happen. [2] show that a trained critic can actually be used to perform rejection sampling on a trained generator and significantly improve the performance of the trained generator. Thus, as training progresses, the critic can serve as a useful arbiter of which samples are 'good'. Then, if we accept the premise that updating on 'good' samples improves GAN training, we should be able to use the critic during training to make decisions about which samples to update on. But why should we accept this premise? Why would updating on the 'bad' samples hurt instead of helping? In this work, we provide a partial answer by showing that in practice, gradient updates derived from samples the critic deems 'bad' can actually point *away* from the true data manifold.

Since the critic's ability to tell us which samples are bad improves during training, we anneal the fraction of the batch that is used for updates as training progresses. In the beginning of training, we use samples from the entire batch, and gradually reduce $k$ after each training epoch.

Our contributions can be summarized as follows:

- We propose a simple 'one-line' modification to the standard GAN training algorithm that only updates the generator parameters on the samples from the mini-batch that the critic scores as most realistic.

- We thoroughly study (on a 'toy' dataset) the mechanism by which our proposed method improves performance and discover that gradients computed on discarded samples would point in the 'wrong' direction.

- We conduct further experiments on the CIFAR [23] and ImageNet [40] datasets and show that our proposed modification can significantly improve FID [18] numbers for several popular GAN variants.

## 2   Background

**Generative Adversarial Networks:**   A Generative Adversarial Network (GAN) is composed of a generator, $G$, and a critic, $D$, where in practice both $G$ and $D$ are neural networks. The generator takes as input a sample $z$ from a simple prior distribution $p(z)$ and is trained so that its output appears indistinguishable from a sample from the target distribution $p(x)$. The critic is trained to be able to *discriminate* whether a sample is from the target distribution, $p(x)$ or from the generator's output distribution $G(z), z \sim p(z)$. Both networks are trained via a min-max game $\min_G \max_D V(D, G)$ where $V(D, G)$ is a loss function. For example, as originally proposed in [15], $V(D, G) = \mathbb{E}_{x \sim p(x)}[\log D(x)] + \mathbb{E}_{z \sim p(z)}[\log(1 - D(G(z)))]$. Many alternate formulations of $V(D, G)$ have been proposed; for a survey see [25]. In practice, mini-batches of $B$ samples $\mathcal{X} = \{x_i \sim p(x), i = 1, \dots, B\}$ and $\mathcal{Z} = \{G(z_i), z_i \sim p(z), i = 1, \dots, B\}$ are used in alternating

stochastic gradient descent to relax the minimax game:

$$\theta_D \leftarrow \theta_D + \alpha_D \sum_{\mathcal{X}} \nabla_{\theta_D} V(D, G) \tag{2.1}$$

$$\theta_G \leftarrow \theta_G - \alpha_G \sum_{\mathcal{Z}} \nabla_{\theta_G} V(D, G) \tag{2.2}$$

where $\alpha_D$ ($\theta_D$) and $\alpha_G$ ($\theta_G$) are the learning rates (and parameters) for the critic and generator respectively. Intuitively, the generator is trained to "trick" the critic into being unable to correctly classify the samples by their true output distributions.

**Top-$k$ Operation:** The top-$k$ operation does what its name suggests: given a collection of scalar values, it retains only the $k$ elements of that collection that have the highest value. We use $\max_k\{Q\}$ to denote the largest $k$ elements from a set $Q$ of scalars.

## 3   Top-$k$ Training of GANs

### 3.1   The Proposed Method

We propose a simple modification to the GAN training procedure. When we update the generator parameters on a mini-batch of generated samples, we simply zero out the gradients from the elements of the mini-batch corresponding to the lowest critic outputs. More formally, we modify the generator update step from Equation 2.2 to

$$\theta_G \leftarrow \theta_G - \alpha_G \sum_{\max_k\{D(\mathcal{Z})\}} \nabla_{\theta_G} V(D, G)$$

where $D(\mathcal{Z})$ is shorthand for the critic's output for all entries in the mini-batch $\mathcal{Z}$. Intuitively, as training progresses, the critic, $D$, can be seen as a scoring function for the generated samples: a generated sample that is close to the target distribution will receive a higher score, and a sample that is far from the target distribution will receive a lower score. By performing the top-$k$ operation on the critic predictions, we are only updating the generator on the 'best' generated samples in a given batch, as scored by the critic. A diagram of our approach is shown in Fig. 1.

### 3.2   Annealing $k$

Early on in training, the critic may not be a reliable scoring function for samples from the generator. Thus, it won't be helpful to throw out gradients from samples scored poorly by the critic at the beginning of training – it would just amount to throwing out random samples, which be roughly equivalent to simply using a smaller batch size.

Thus, we set $k = B$, where $B$ is the full batch size, at the start of training and gradually reduce it over the course of training. In practice, we decay $k$ by a constant factor, $\gamma$, every epoch of training to a minimum of $k = \nu$. We use the minimum value $\nu$ so that training doesn't progress to the point of only having one element in the mini-batch. Refer to Section 4 and 5 for more details on the values of $\gamma$ and $\nu$ we used in practice.

**Top-$k$ Training of GANs in PyTorch**   A sample PyTorch-like [37] code snippet is available below. In the code snippet, `generator_loss` represents any standard generator loss function. Our top-$k$ GAN training formulation amounts simply to the addition of line 8 of the example code. This highlights its ease of implementation and generality.

```
1  # Generate samples from the generator
2  fake_samples = Generator(prior_samples)
3
4  # Get critic predictions
5  predictions = Critic(fake_samples)
6
7  #Get topk predictions
8  topk_predictions = torch.topk(predictions, k)
9
10 # Compute loss for generator on top-k predictions
11 loss = generator_loss(topk_predictions)
```

| Number of Modes | % High Quality Samples (GAN) | % High Quality Samples (Top-$k$) | % Modes Recovered (GAN) | % Modes Recovered (Top-$k$) |
|---|---|---|---|---|
| 25 | 85.6 | **95.5** | **100** | **100** |
| 64 | 73.8 | **81.8** | 96.2 | **100** |
| 100 | 40.3 | **54.7** | 94.6 | **100** |

Table 1: GAN training with and without Top-$k$ on a Mixture of Gaussians. 'High Quality Samples' measures the fraction of samples that lie at most 4 standard deviations away from the nearest mode. 'Modes Recovered' measures the fraction of modes which have at least one high quality sample.

## 4 Mixture of Gaussians

In this section we investigate the performance of top-$k$ GAN training on a toy task in order to better understand its behaviour. Following [2] our toy task has a target distribution that is a mixture of Gaussians with a varying number of modes. We will first demonstrate and discuss how top-$k$ training of GANs can reduce mode dropping (i.e. learning to generate only a subset of the individual mixture components) and boost sample quality in this setting. We then move on to discuss an interesting phenomenon: when gradient updates are performed on the bottom-$k$ instead of the top-$k$ batch elements, samples actually tend to be pushed away from their nearest mode. This phenomenon suggests a mechanism by which top-$k$ training improves GAN performance: it doesn't use these "unhelpful" gradients in its stochastic mini-batch estimate of the full gradient.

### 4.1 Experimental Setup

We follow the same experimental setup as in [2] and [41]. We set the target distribution to be a mixture of 2D isotropic Gaussians with a constant standard deviation of 0.05 and means evenly spaced on a 2D grid. The generator and critic are 4-layer MLPs with 256 hidden units in each layer, which are trained using the 'non-saturating' loss from [15]. We train each network with a constant batch-size of 256 for all experiments. All networks are trained with Adam optimizer with a learning rate of $10^{-4}$ [22].

For all experiments we measure (as in [2]): $i$) High quality samples: percent of samples that lie at most 4 standard deviations away from the nearest mode and $ii$) Modes recovered: percent of modes which have at least one high quality sample. The more modes the generator is able to recover, the less we say it suffers from mode-dropping. For this evaluation, we randomly sample 10,000 samples from the generator. We train the networks for 100,000 iterations and decay $k$ every 2,000 iterations.

For top-$k$ training, we initialize $k$ to be the full mini-batch size. We use a decay factor, $\gamma$, of 0.99 to decay $k$ until it reaches its minimum value, $\nu$, of 75% of the initial mini-batch size, or 192. Formally, we do: $k \leftarrow \max(\gamma k, \nu)$

**Quantitative Results** The quantitative results for all Mixture-of-Gaussians experiments are summarized in the Appendix. We see that as we increase the number of modes in the target distribution, top-$k$ training is able to improve both the fraction of modes recovered and the fraction of high quality samples: As the number of modes is increased from 25 to 100, the number of high-quality samples decreases dramatically for normal GANs; top-$k$ training performs significantly better. The fact that the number of modes recovered by performing top-$k$ training is larger than the number recovered without shows that top-$k$ training may help mitigate mode-dropping.

As GAN training progresses, the critic implicitly learns to classify whether or not a sample is drawn from the true distribution. Thus, generated samples that are in-between the modes of the target distribution tend to yield lower outputs from the critic [2]. By discarding these samples, we focus updates to the generator parameters on the best-scoring samples in the mini-batch, which results in better GAN training results. But why does this happen? In the next section, we will show that gradient updates computed on samples which are 'in-between modes' (samples that top-$k$ sampling will discard) often cause samples to move in the *wrong* direction (i.e. away from the nearest mode) after each gradient update.

## 4.2 Why Does Throwing out Bad Samples Help?

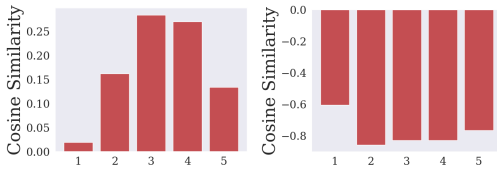

Figure 2: Cosine similarity between the direction moved by a generator sample after an update to the direction to the nearest mode for top-$k$ (left) and bottom-$k$ (right) samples. Each bin in the histogram represents samples which are within a given standard deviation away from the nearest mode.

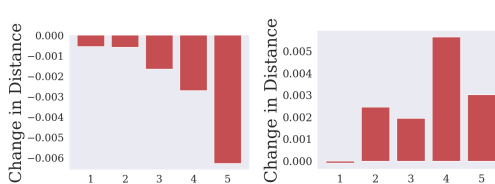

Figure 3: Change in distance to the nearest mode for generator samples after an update for top-$k$ (left) and bottom-$k$ (right). Each bin in the histogram represents samples which are within a given standard deviation away from the nearest mode.

In this section, we examine what happens when the GAN generator is updated on either the best-scoring elements or *worst*-scoring elements in a mini-batch. This sheds some light on a possible reason that top-$k$ training is helpful: gradient updates computed on the worst-scoring samples tend to move samples away from the nearest mode.

For this experiment, we train a normal GAN for 50,000 iterations (half the number of iterations as in the experiments from Table 1) on a mixture of 25 Gaussians. Besides halving the number of iterations, we keep the settings otherwise the same as in Table 1. We then draw 1,000 samples from the generator's prior distribution $z \sim p(z)$. We use this batch $z$ of 1,000 samples to generate samples from the generator. For each of those samples, we compute the direction to the nearest mode, which we refer to as the *oracle direction*[2].

Then, with these oracle directions as a reference point, we compare top-$k$ and 'bottom-$k$' updating, which respectively update the generator using only the top-7,500 or bottom-2,500 critic predictions. After performing a gradient descent step, we re-compute the generator samples using the same $z$ that were used to produce the oracle directions. We then measure the movement of the samples after the update steps. By isolating the effect of one gradient step, we can understand what happens when the generator is updated using the 'bad' samples compared to what happens when it is updated using the 'good' samples. This comparison is unbiased because we use the same generator and critic and the same batch of $z$ for both the top-$k$ and the bottom-$k$ update.

In order to understand why updating on the worst-scoring samples is harmful, we evaluate the cosine similarity between the oracle direction and the displacement computed above, for each sample. By evaluating the cosine similarity,

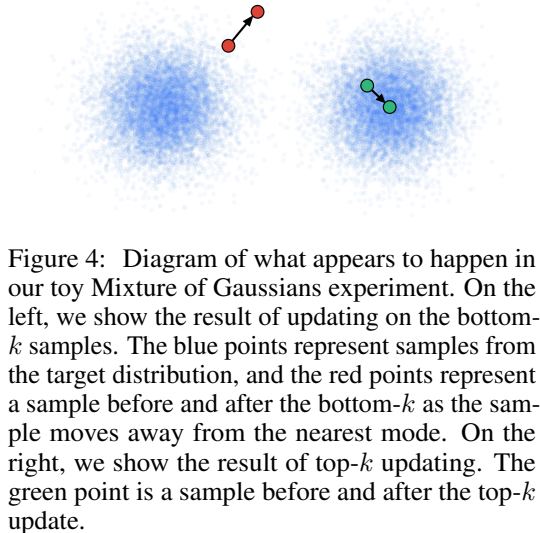

Figure 4: Diagram of what appears to happen in our toy Mixture of Gaussians experiment. On the left, we show the result of updating on the bottom-$k$ samples. The blue points represent samples from the target distribution, and the red points represent a sample before and after the bottom-$k$ as the sample moves away from the nearest mode. On the right, we show the result of top-$k$ updating. The green point is a sample before and after the top-$k$ update.

we are loosely measuring the quality of the gradient update: Roughly speaking, the closer the cosine similarity is to 1, the better the update is, since a value of 1 means that the points are being pushed in the exact direction of the mode. The closer the cosine similarity is to -1, the worse the gradients are, since a value of -1 means that points are being pushed in the opposite direction of the nearest mode. The results of this experiment are summarized in Fig. 2: We bin points by how many standard deviations away they are from the nearest mode and compute a histogram of mean cosine distance for

|  | 1 | 2 | 3 | 4 | 5+ |
|---|---|---|---|---|---|
| GAN | 40.8 | 26.3 | 12.4 | 7.8 | 12.7 |
| Top-$k$ GAN | 74.9 | 14.7 | 2.8 | 2.3 | 5.3 |
| Target data | 68.2 | 27.2 | 4.2 | 0.2 | 0.1 |

Table 2: Percent of samples between $n-1$ and $n$ standard deviations away from the nearest mode. The "Target data" represents what percent of points are within each mode for a Gaussian distribution, and therefore serves as the target. We Top-$k$ sampling reproduces the underlying distribution much more faithfully.

| DC-GAN | Top-$k$ + DC-GAN | WGAN+GC | Top-$k$ + WGAN+GC | WGAN+GP | Top-$k$ + WGAN+GP |
|---|---|---|---|---|---|
| $38.09 \pm 0.3$ | $\mathbf{35.62 \pm 0.4}$ | $37.33 \pm 0.3$ | $\mathbf{34.41 \pm 0.3}$ | $31.80 \pm 0.2$ | $\mathbf{29.83 \pm 0.2}$ |

| MS-GAN | Top-$k$ + MS-GAN | SN-GAN | Top-$k$ + SN-GAN | SA-GAN | Top-$k$ + SA-GAN |
|---|---|---|---|---|---|
| $27.33 \pm 0.2$ | $\mathbf{26.54 \pm 0.3}$ | $21.36 \pm 0.2$ | $\mathbf{19.80 \pm 0.2}$ | $19.02 \pm 0.2$ | $\mathbf{17.93 \pm 0.2}$ |

Table 3: Reporting the FID-50k metric on the CIFAR dataset for various GAN architectures. The GAN architectures considered are DC-GAN, WGAN with Gradient Clipping, WGAN with Gradient Penalty, Mode-Seeking GAN, Spectral-Normalization GAN and Self-Attention GAN.

each bin. The '5' bar in each plot are all the points that greater than 4 standard deviations away (not high-quality samples).

This experiment gives a somewhat surprising result when only the bottom-$k$ update is performed: The cosine similarity between the update direction and the oracle direction is *negative* in this case, even for samples that are we consider to be high-quality samples (those within 4 standard deviations from the closest mode). This suggests that points are being actively pushed away from the mode that they are already close to. For samples which are very close to the nearest mode, the cosine similarity is less important since these samples are already "good'. That is what we see when we update the generator using the proposed top-$k$ method. The points that are more than 4 standard deviations away move in the correct direction, even when the generator was not directly updated on those exact points because of the masking operation from top-$k$. Figure 4 further visualizes this behavior.

We also compute the change in distance to the nearest mode after the gradient update is done. We want the distance to the nearest mode to decrease after the gradient update which means that the generated distribution is getting closer to the target distribution. We notice similar effects to cosine similarity, where when updating only on the bottom-$k$ samples, we see that the distance to the nearest mode increases after the gradient update, while updating only on the top-$k$ samples, the distance tends to decrease. The top-$k$ plot shows that the further the point is from the nearest mode, the more it moves closer to it, and the points that are already very close to the mode remain relatively unaffected by the gradient update. This experiment further shows how the bottom-$k$ samples actively result in a *worse* generator after an update. Finding the mean gradient signal from the full batch will result in the added negative influence from the bottom-$k$ samples; our method reduces the negative effects by simply discarding the bottom-$k$ samples, which is a computationally efficient, effective and easy-to-implement solution.

We also investigate what percent of points lands within a given standard deviation from the nearest mode. Ideally, since the underlying distribution is a mixture of i.i.d Gaussians, the generated distribution should resemble the "Target" distribution for each standard deviation. We tabulate the results in Table 2, where we see that using top-$k$ sampling for GANs, we are able to more faithfully recover the target distribution. A vanilla GAN smears the distribution, as the generated distribution has a long tail, but using top-$k$ sampling, we are able to perform significantly better on the recovering the "true" target distribution.

| SAGAN | Top-$k$ + SAGAN |
|:---:|:---:|
| 19.98 | **18.44** |

Table 4: FID for SAGAN on ImageNet.

| Model | Vanilla | + Top-$k$ |
|:---:|:---:|:---:|
| BigGAN [4] | 14.73 | **13.34** |
| ICR-BigGAN [51] | 9.21 | **8.57** |

Table 5: Official FID for BigGAN and ICR-BigGAN on CIFAR-10. The value in blue represents the official state-of-the-art value.

## 5  Experiments on Image Datasets

In order to investigate whether top-$k$ training scales beyond toy tasks, we apply it to several common GAN benchmarks.

**Experiments on CIFAR-10**   Since the most common application of GANs is to image synthesis, we exhaustively evaluate our method on different GAN variants using the CIFAR-10 dataset [23]. The CIFAR dataset is a natural image dataset consisting of 50,000 training samples and 10,000 test samples from 10 possible classes and is probably the most widely-studied GAN benchmark. For all of our experiments, we compute the FID [18] of the generator using 50,000 training images and 50,000 generator samples. It is important to note that we use a PyTorch Inception [37] network to compute the FID, instead of the TensorFlow implementation [37]. This means that the overall values will be lower, but it does not affect relative ranking of models, so it still enables unbiased comparisons. Since we use the same implementation to compute each FID value in this paper, the results are comparable. A short explanation of the different GAN variants is available in the Appendix.

For all experiments, we use a mini-batch size of 128 and initialize the value of $k$ to be the full mini-batch size. Unless otherwise noted, we set $\gamma$ to 0.99, where we decay $k$ after every epoch until it reaches the value of half the original batch size, $\nu = 64$. We considered values of $\gamma$ in the range of $[0.75, 0.999]$ and values of $\nu$ in the range of $[32, 100]$. **For each model, all other hyperparameters used were same as those used in the paper proposing that model.** By leaving the original hyperparameters fixed, we can demonstrate how top-$k$ training is a "drop-in" improvement for each of these GANs.

The results of these experiments are summarized in Table 3. We see that using top-$k$ sampling significantly helps the performance across all GAN variants. For the simpler GAN variants, such as DCGAN and WGAN with gradient clipping, we see that the performance is significantly better when using top-$k$. Even for the state-of-the-art GAN architectures, such as Self-Attention GAN and Spectral Normalization GAN, our method is able to outperform the baseline by a good margin. We speculate that we achieve larger improvements on less sophisticated GAN models for the simple reason that there is less room for improvement on the more sophisticated models (though top-$k$ training yields substantial improvements in all cases).

To investigate the effect of Top-$k$ sampling on reduce mode-collapse in GANs, we compute the standard *Number of statistically Distinct Bins* (NDB/$K$) metric, in which a lower score is better [39]. Using a SAGAN model baseline on CIFAR-10 with a standard $K = 100$, top-$k$ sampling improves the NDB results from a basline GAN score of 0.75 to 0.60 ($K = 100$ refers to the NDB/$K$ metric, not top-$k$ GAN).

**Experiments On ImageNet**   ImageNet [40] is a large-scale image dataset consisting of over 1.2 million images from 1,000 different classes. Training a GAN to perform Conditional Image Synthesis on the ImageNet dataset is now a standard GAN benchmark to show how a given GAN scales. Since this benchmark is considered more difficult than training an image synthesizer on the CIFAR-10 dataset, we include these experiments as evidence that top-$k$ training can scale up to more difficult problem settings.

We run our experiments with the Self Attention GAN (SAGAN) [47], since it is relatively standard, has open-source code available, and is easy to modify. As in our CIFAR-10 experiments, we train the baseline model using the same hyper-parameters as suggested in the original paper. We set the top-$k$ decay rate – $\gamma$ — to 0.98 due to the large size of the dataset. We report the FID score on 50,000 generated samples from the generator. The results are summarized in Table 4.

**BigGAN** For conditional BigGAN [4] and ICR-BigGAN [51], we set batch size to 256 and train for 100k steps. We use the original hyperparameters of these models and apply Top-$k$ training in addition. We used $\gamma = 0.999$, $\nu = 0.5$ with annealing every 2000 steps due of the larger batch-size used. We are able to improve upon both models and achieve new state-of-the-art GAN results for CIFAR-10, as shown in Table 5. This further shows how Top-$k$ sampling can improve state-of-the-art GAN variants which consequently results in a new state-of-the-art in image synthesis on CIFAR-10. Although the performance difference may appear to be small, using Top-$k$ sampling we approach significantly closer to a potential best value attainable for GANs on CIFAR-10.

Broadly speaking, the results show that top-$k$ training can substantially improve FID scores, despite being an extremely simple intervention. The fact that we were able to achieve state-of-the-art results with minimal hyper-parameter modifications is a testament to the broad applicability of the top-$k$ training technique.

**Anomaly Detection** We first investigate its utility as a general tool for GAN-based architectures by testing it on anomaly detection [5], finding that it improves results substantially. The experimental settings and results are expanded in the Appendix. We then conduct experiments on the CIFAR-10 dataset [23] using six different popular GAN variants and on the ImageNet dataset [40] using the SAGAN [47] architecture. We test on a variety of GAN variants to ensure that our technique is generally applicable and the quantitative results indicate that it is: top-$k$ training improved performance in all of the contexts where we tested it.

| SAGAN ($\rho$=64) | Top-$k$ SAGAN ($\rho$=64) | SAGAN ($\rho$=128) | Top-$k$ SAGAN ($\rho$=128) | SAGAN ($\rho$=256) | Top-$k$ SAGAN ($\rho$=256) |
|---|---|---|---|---|---|
| 21.1 | **19.8** | 19.0 | **17.9** | 18.6 | **17.4** |

Table 6: The effect of batch-size and Top-$k$ sampling, where $\rho$ represents the batch-size. Top-$k$ sampling is added to a SAGAN on CIFAR-10, and is effective for different batch-sizes for GAN training.

**Effect of Batch-Size** Since batch-size has been shown to be a critical factor in GAN training [41, 4], we train a SAGAN model on the CIFAR-10 dataset for different batch-size with and without top-$k$ sampling. Using the same hyperparameters for training, we report the results in Table 6. We see that as the batch-size $\rho$ of the is increased from 64 to 256, top-$k$ sampling continues to outperform the baseline model by a significant margin. This further shows the usefulness of top-$k$ sampling, as it is able to improve GAN training over varying batch-sizes, without changing any hyperparameters.

| $\gamma = 0.999$ | $\gamma = 0.99$ | $\gamma = 0.9$ | $\gamma = 0.5$ | $\nu = 0.9$ | $\nu = 0.5$ | D | G & D |
|---|---|---|---|---|---|---|---|
| 18.68 | **17.93** | 18.14 | 25.30 | 18.47 | **17.93** | 27.44 | 27.57 |

Table 7: FID scores for SAGAN on CIFAR over a range of ablation studies. For each experiment, all other hyperparameters are as proposed. Note: $\nu$ is listed as a percent of full mini-batch size (128). The **bold** values represent the proposed values of the given hyperparameters. Experiment labeled "D" represents applying top-$k$ on just the critic. Experiment labeled "G & D"represents applying top-$k$ on both the generator and critic.

**Examining the Main Hyper-parameters** In this section we study the effects of the various hyper-parameters involved in performing top-$k$ GAN training. In particular, we focus on the effect of the decay rate, $\gamma$; the minimum value of $k$, $\nu$; and the effect of applying top-$k$ updates to the critic as well as just to the generator. We train a SAGAN [47] on CIFAR-10 dataset [23]. The results are presented in Table 7.

The first thing to notice is that using too small of a value of $\gamma$ hurts performance by discarding too many samples early on in training. Secondly, using too large a value for $\nu$ degrades performance, because if $\nu$ is too large, then too few samples are discarded, and top-$k$ training becomes similar to normal training.

# 6   Related Work

**Generative Adversarial Networks**   Recent GAN reserach has focused on generating increasingly realistic images. Both [34] and [25] serve as good overviews on the state of current GAN research. A wide vareity of techniques have been proposed to improve GAN training, including mimicking or using large batches [41, 4], different GAN architectures [6, 47, 38, 4, 28], and different GAN training objectives [30, 1, 33, 49, 3, 31, 9, 52]. Alternatively, variance reduction has been explored as a way to stabilize the GAN training procedure: [14] proposes solve a variational-inequality problem instead of the solving the min-max two player GAN objective, and [6, 7] propose using an extra-gradient method while training. Some recently proposed methods have tried to improve GANs from a computer graphics lens [21, 20, 19]. Other work focuses on conditional image synthesis on large-scale datasets such as ImageNet [40], [35, 32, 4, 8, 44, 49]. Some work even focuses on totally different ways to evaluate generative models (and GANs in particular) [36, 17].

**Effectively Using critic Outputs**   Our work is more closely related to the line of research which aims to use the critic output to further augment the GAN training procedure. The goal is to distill more information from the critic than is possible using only standard GAN optimization techniques. [2] proposed a post-training procedure to use rejection sampling on the critic outputs for the generated samples. [44] show that a similar trick can work *during training*, by only updating the generator using draws from the prior that have themselves been modified in response to the critic output using a gradient correction. [42] shows an effective technique to use the Discriminator's output scores to importance weights the generator loss.

# 7   Conclusion

We have described a technique that is very simple – it requires changing only one line of code – that yields non-trivial improvements across a wide variety of GAN architectures. In fact, it yielded substantial improvements in every context in which we evaluated it. A more sophisticated technique could probably yield slightly better results after substantial tweaking, but there are serious barriers to using such a technique in practice – simplicity tends to win out. We hope that this technique will become standard.

We have also discovered and studied an interesting phenomenon in the Mixture-of-Gaussians setting: generators updated using top-$k$ updating push samples toward their nearest mode, while generators updated using bottom-$k$ updating tend to push samples *away from their nearest mode*. This partially explains why top-$k$ sampling is successful (it removes from the mini-batch incorrect contributions to the estimate of the gradient), but it is also interesting in its own right. We hope that further study of this phenomenon can spur advances in our understanding of the GAN training procedure: perhaps it can connected with other interesting experimental observations about GANs, or used to explain performance improvements from other heuristically motivated techniques.

## Broader Impact

In this paper we present a simple yet effective GAN training technique, which significantly improves the performance of many GAN variant while adding almost no additional training cost. This method can be useful for any practical application where GAN training can be useful, such as Computer Graphics. It can be used to boost GAN performance, and therefore help artists and content creators with their designs and creations. Our technique can also be useful when there is only a small amount of training data available for training, as training GANs on the data can help generate synthetic data, which the model can then use to train.

## Acknowledgements

We would like to thank Nvidia for donating DGX-1 and Compute Canada for providing resources for this paper. We would also like to thank Roger Grosse and Vaishnavh Nagarajan for some insightful discussions towards this work.

## Footnotes

* Samarth Sinha and Zhengli Zhao contributed equally as joint first authors.

[2] Note that, if some modes are very over-represented, the oracle direction won't be quite right, but in practice this is not a big issue.

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
