[Supplementary Material]

# A   GAN Variants

We apply top-$k$ training to all of the following GAN variants:

- DC-GAN [38]: A simple, widely used architecture that uses convolutions and deconvolutions for the critic and the generator.
- WGAN with Gradient Clipping [1]: Attempts to use an approximate Wasserstein distance as the critic loss by clipping the weights of the critic to bound the gradient of the critic with respect to its inputs.
- WGAN with Gradient Penalty [16]: Improves on WGAN [1] by adding an gradient norm penalty to the critic instead of clipping weights.
- Mode-Seeking GAN [29]: Attempts to generate more diverse images by selecting more samples from under-represented modes of the target distribution.
- Spectral Normalization GAN [32]: Replaces the gradient penalty with a (loose) bound on the spectral norm of the weight matrices of the critic.
- Self-Attention GAN [47]: Applies self-attention on both the generator and critic.

# B   Anomaly Detection

| Held-out Digit | Bi-GAN | MEG | Top-$k$+MEG |
|:--------------:|:------:|:-----:|:----------:|
| 1 | 0.287 | 0.281 | **0.320** |
| 4 | 0.443 | 0.401 | **0.478** |
| 5 | 0.514 | 0.402 | **0.561** |
| 7 | 0.347 | 0.29 | **0.358** |
| 9 | 0.307 | 0.342 | **0.367** |

Table 8: Experiments with Anomaly Detection on MNIST dataset. The 'Held-out Digit' is the digit that was held out of the training set during training and treated as the 'anomaly' class. The numbers reported is the area under the precision-recall curve.

We conduct experiments on the anomaly detection task and model from [24] (as also used in [41]). We augment Maximum Entropy Generators (MEG) with top-$k$ sampling on the generator. MEG performs anomaly detection by learning the manifold of the true distribution; by learning a better generator function, MEG aims to be able to learn a better model for anomaly detection.

As in [24], we train a generative model on 9 out of the 10 digits on the MNIST dataset [26], where the images of the held-out digit are meant to simulate the anomalous examples that the method is intended to find. Since the results from [24] using MEGs are comparable to those from [46] (which uses BiGANs [12]) we report both MEG and BiGAN-based solutions as our baseline methods. The results, which are shown in 8, are reported in terms of area under the precision-recall curve, as in [24]. Broadly speaking, they show that applying top-$k$ training to the MEG-based method yields results better than both the MEG-based and BiGAN-based methods for all 5 of the held-out digits we tried. Though we only apply top-$k$ training to the MEG method in this instance, we suspect it can be fruitfully applied to BiGAN-based methods as well. By applying top-$k$ training to a task other than image synthesis, we aim to show that it is a generally useful technique, rather than a task-specific hack.

# C   Applying Top-$k$ Updates to The Critic Hurts Performance

Perhaps most interesting of all is that applying top-$k$ updates to the critic (instead of just to the generator, as we do in all other experiments) completely destabilizes training. Further study of this phenomenon is best deferred to future work, but we can briefly speculate that modifying the critic in this way is harmful because it causes the critic only to update for modes that the generator has learned early on, ignoring other parts of the target distribution and thus preventing the generator from learning those parts.

# D  Fréchet Inception Distance:

[18] proposed Fréchet Inception Distance (FID) as a metric to measure how well a generative model has fit an target distribution. The metric utilizes an internal representation from a pre-trained Inception classifier [43] and measures the Fréchet distance from the target distribution $p(x)$ to the generated distribution $G(z)$ [13]. The FID score is calculated by:

$$||m - m_w||_2^2 + Tr(C + C_w - 2(CC_w)^{1/2})$$

where $m$ and $C$ are the mean and co-variances of the Inception embeddings for real-data, and $m_w$ and $C_w$ are the mean and covariance matrix of the Inception embeddings for the generated samples. In practice, 50,000 generated samples are used to measure the FID of a GAN.