[Reviews · NeurIPS 2020]

Review 1

Summary and Contributions: This work proposes a simple technique to improve the quality of the Generator in GAN by only using the most realistic samples based on the Discriminator’s predictions. In particular, the work proposes to discard the gradient contributions from samples which the discriminator predicted least realistic, hence enforcing the generator to only learn from the samples which are most realistic. Experiments are conducted to illustrate that less realistic samples negatively affect the Generator performance by pushing samples to move away from the actual distribution.

Strengths: - A simple and effective technique to improve generation quality of a GAN - Simplicity of the method enables it to be utilized with many existing GAN variation with very minimal modifications - Proposed technique is clearly presented in the paper in an easy to follow manner - Several experiments are provided to support the claims mentioned in the paper

Weaknesses: *Post rebuttal* The authors have provided new data to address my concern. I have adjusted my rating. ================================ - The proposed technique to zero out gradient from the least realistic is an interesting engineering trick. But there is not much to learn from this work intellectually - Empirical evaluation can be improved to support claims provided in the paper. The Mixture of Gaussians experiment provides some support. But to strengthen the soundness of the findings, it is suggested to provide qualitative experiments on CIFAR or ImageNet examples, especially on the impact top-k on mode collapsing problem compared to baselines.

Correctness: yes

Clarity: Proposed technique is clearly presented

Relation to Prior Work: yes

Reproducibility: Yes

Additional Feedback: Some suggestions to improve the manuscript - Better to move the code snippet to Appendix section as the equations already explain the proposed approach. Framework wise implementation can be considered as an additional support. - L34-44 section highlights some important arguments. Improving the quality of it taking a more technical based writing approach with comparison to existing work will improve the manuscript


Review 2

Summary and Contributions: The paper proposes a simple method to improve the training of GAN that leverages the critics' outputs to filter out “unrealistic” samples and use only top-k realistic samples to update the generator. The paper also shows the observation on toy example (Mixture of Gaussian) that top “unrealistic” samples, examples tend to push away from the nearest modes. The experiments show that the method can be easily applied for various GAN models and improve over the baseline on CIFAR-10 and ImageNet datasets. The paper also apply the proposed method for the anomaly detection task on MNIST dataset and show the improvement on one baseline model. The paper is well-written and the method is simple to understand. However, I have several concerns about the paper. Firstly, what I'm concerned the most is the novelty of the proposed method as the key idea of rejection sampling using critics has been discussed in previous methods, e.g., [2], [38]. The method proposes the incremental changes by hard importance re-weighting the contributions of the synthetic samples (1: for high-score and 0 for low scores) based on the number k. The choice of k is rather heuristic by decaying with a scaling factor. Secondly, the problem (e.g., the mode collapse, gradient vanishing?) aimed to be addressed in the paper is unclear to me as the authors only show the one observation on the gradients at the half of number iterations on a toy example. Gradients may be different at the beginning and the ending at the training should be observed as well. Thirdly, rejecting the low-density samples at the end may lead to the generator only focus on generating the high-density samples. Eventually, it may affect the diversity and hard to model, e.g., long-tail distributions in practice. Finally, no comparison to related works, e.g., [2]. I have some questions about these concerns below and want to hear authors in the rebuttal.

Strengths: 1. The simple method that is easy to be integrated into GAN models to improve the baseline. 2. The paper is clear and well-written. 3. Improvements over the number of GAN baseline models.

Weaknesses: 1. The novelty of the method is rather incremental, and heuristic in the selecting k number. 2. The problem of GAN to be addressed in the paper is unclear. 3. No comparison of similar works using critics for rejection sampling, e.g., [2].

Correctness: Yes, they sounds correct.

Clarity: The paper is well-written and clear to read.

Relation to Prior Work: The most related work is [2], the paper is incremental improved from the hypothesis of [2]. But, there is no comparison with this work to see the trade-off between the training time and the performance.

Reproducibility: Yes

Additional Feedback: == Post rebuttal == I thank the authors for the response. I agree with all reviewers that the method is simple, effective, and generalized to improve GAN methods. However, I think the rebuttal does not address well all my concerns, some of them remain: 1. Regarding the novelty of the paper, using the D scores as the feedback to improve the generator quality is not new. A number of methods has been proposed e.g., [2], [38], [*] [**]. What is new in the paper is the simple way how to use the discriminator scores. However, quite missing substantial discussion and comparison to related works to understand the advantages of the proposed method over the existing works, e.g., stronger improvements or better training time, etc? [*] Metropolis-Hastings Generative Adversarial Networks [**] Your GAN is Secretly an Energy-based Model and You Should use Discriminator Driven Latent Sampling 2. The inconsistency of \nu value in the experiment is not addressed in the rebuttal. This is doubt about the simplicity of the method, in fact, likely needs to be tuned quite a lot of hyper-parameters (k, \gamma, \nu, …). Furthermore, decaying k with some parameters seems to look a trick and may highly depend on dataset size, and the number of training iterations. 3. The analysis of what improved GAN is not solid and insufficient and thus needs more studies. Hence, I keep my rating. =============== Although rejecting the bad samples are empirically helpful, I'm concerned about the diversity of generated samples. For example, on the toy example, the way of evaluating high-quality samples cannot evaluate the diversity of samples for each mode. I suggest computing std on of high-quality samples for each mode to see how diverse the samples are. Can the author show the gradients at roughly the beginning and the end at the end of the training, also the final distribution learned by generator compared to the data distribution? The value of values of \nu in the range of [32; 100] which is not consistent with \nu in Table 5 represents the percent of the full mini-batch size. Moreover, if accepting that \nu is represented by percent of full mini-batch size (128), its maximum value is 100 / 128 = 0.78125. How \nu can be 0.9 in Table 5? The chosen of decay number k is rather heuristic. Is the selected optimal value (e.g., Table 5 on CIFAR with SAGAN) the same for other methods and datasets, and also when the batch size and number of training iterations are changed? The paper could certainly be improved if being quantitatively compared to the related work, e.g. [2] for FID and training time. Line 262 – 268: I think there is a mistake to put discussion on CIFAR-10 and ImageNet for anomaly detection? -- suggestion -- Rejecting the bad samples in other words means that assigning the zero-weights the loss on these samples and one-weights for others. Perhaps the author may think a more general way, e.g., assigning different weights for different synthetic samples.


Review 3

Summary and Contributions: This paper proposed to only updates the generator parameters on the samples from the mini-batch that the critic scores as most realistic. Such a training way is proved to be able to improve the generation quality (in terms of FiD) for several popular GAN models. The authors studied and analyzed this mechanism on a toy dataset, and they found that gradients computed on discarded samples would point in the ‘wrong’ direction.

Strengths: 1. The proposed top-k training of GANs is interesting and inspiring. Moreover, it is effective in improving generation quality. 2. The analysis on the toy dataset makes sense, which provides some evidence that "bad" samples should be thrown away during the optimization.

Weaknesses: ###post rebuttal### The authors' feedback partly addressed my concerns. While I still think the analysis is supposed to be more solid, and I agree with R2 that "the analysis of what improved GAN is not solid and insufficient and thus needs more studies." ################## 1. The analysis is not very solid. This paper actually discussed how to deal with "hard cases". There are two common methods, when the training is hard to converge, abandoning hard cases can make the training more stable, while when the task is easy to converge, paying more attention to hard cases to achieve better performance. As a result, the "bottom-k" samples may play different roles during the whole training process, while the authors only conduct analysis on the setting of half the number of iterations, it would be better to conduct experiments in different period. 2. It would be better to further discuss how different annealing strategies influence the performance. I think this is an important ablation study to verify the claim "improving GAN performance by throwing away bad samples".

Correctness: Yes.

Clarity: Yes.

Relation to Prior Work: Yes.

Reproducibility: Yes

Additional Feedback:


Review 4

Summary and Contributions: This paper proposes a simple modification to the GAN training algorithm. This modification is about how to update the generator parameters. In the normal way, the generator parameters are updated on all samples. Instead of using all these samples, this paper throws away the samples that are scored as least realistic and updates the generator parameters on the rest of the samples. This way, it makes the samples at the modes become more stable and improves the performance. They apply this method to BigGAN, train the generator and the discriminator, and improve the FID from 9.21 to 8.57. They also apply this method to various GANs trained on CIFAR and ImageNet datasets and improve the FID on CIFAR-10.

Strengths: The modification is simple and easy to implement. It only adds one line of code. This method is also very general and can be applied to various GANs. The toy example and analysis are helpful for understanding.

Weaknesses: The proposed method improves a lot on CIFAR-10, which is a relatively small dataset. When applied to large dataset like ImageNet, the improvement is not that significant. The dataset size and the batchsize might be important factors of how large gain the model can have. I would expect the authors to add an ablation study of the comparison using different batchsizes. Does the performance gain decreases when the batchsize gets larger? Although the results show that this simple modification improves the performance, it would be nice if the authors can present more convincing results, for example, the comparison with other SOTA methods.

Correctness: Correct

Clarity: The writing is clear.

Relation to Prior Work: In contrastive learning there are many previous works discussing about using hard negatives and nearest neighbors as negative examples. It would be better if the authors can also mention them.

Reproducibility: Yes

Additional Feedback: This paper is a really good example of simple but effective method. I would expect the authors to add an ablation study of the comparison using different batchsize and dataset size. I have read the authors response and other reviewers' comments.

[Author Response · NeurIPS 2020]

Thanks for the reviews! We have run several new experiments in response to reviewer comments. See detailed replies to questions below:

**[R1]** **: impact of top-k on mode collapse compared to baselines.** To show that Top-$k$ sampling reduces mode-collapse in GANs, we compute the standard *Number of statistically Distinct Bins* (NDB/$K$) metric - a lower score on this metric is better. **On CIFAR-10, top-$k$ sampling improves the NDB number from 0.75 to 0.60, using a SAGAN baseline. This suggests that top-$k$ sampling actually reduces mode-collapse.** We'll add this discussion to the paper.

|  | 1 | 2 | 3 | 4 | 5+ |
|---|---|---|---|---|---|
| GAN | 40.8 | 26.3 | 12.4 | 7.8 | 12.7 |
| Top-$k$ GAN | 74.9 | 14.7 | 2.8 | 2.3 | 5.3 |
| Target data | 68.2 | 27.2 | 4.2 | 0.2 | 0.1 |

Table 1: Fraction of samples between 0 and 1 standard deviations away from their closest mode, and between 1 and 2, etc. Top-$k$ sampling reproduces the underlying distribution much more faithfully.

**[R2]** **: The novelty of the method is rather incremental** The main contribution of this method is to present a previously unknown property of GAN training, which significantly hinders GAN training across different GAN architectures and objectives. We also study one reason for the existence of the phenomenon in the Mixture of Gaussian setting, where we show how the samples travel away from the nearest mode, when a gradient update is applied on the *worst* samples in a batch. We also show a significant boost in performance for many state-of-the-art GANs, namely BigGAN and ICRGAN, which further strengthens the importance of the proposed technique.

**[R2]** **: No comparison of similar works using critics for rejection sampling**

We performed new experiments to address this point: First, we perform Discriminator Rejection Sampling (DRS) on a vanilla SAGAN on CIFAR-10. This improves the FID from 19.1 to 18.2, as you'd expect. Then, we performed DRS on a SAGAN trained with top-$k$ updates. This improves the FID from 17.8 to 17.2. **This shows that top-$k$ training and post-processing techniques like DRS are not in competition - you can do both to further improve results!**

**[R3]** **: There are two common methods, when the training is hard to converge, abandoning hard cases can make the training more stable, while when the task is easy to converge, paying more attention to hard cases to achieve better performance.** Please refer to the NDB table above, which suggests that using top-$k$ training does not result in GANs dropping hard-to-model data. The worst samples in a batch are harmful to training, since the Discriminator is unable to provide meaningful gradients, not because the data itself is hard-to-model.

**[R3]** **: how different annealing strategies influence performance.** Table 5 in the submission has this information.

**[R4]** **: Would be nice if the authors can present more convincing results, for example, the comparison with other SOTA methods.** We perform two experiments with models that are the state-of-the-art including BigGAN and ICR-GAN, and are able to get significant improvements on both models. As far as we know, **Our results from Table 4 are the current state-of-the-art for CIFAR-10.**

**[R4]** **: When applied to large dataset like ImageNet, the improvement is not that significant.** Since ImageNet is a very large dataset, there was no hyperparameter tuning done on ImageNet, and the exact same hyperparameters for CIFAR-10 were used for ImageNet. It's likely possible to improve the results even further given more tuning. Also, the improvement in FID from 19.98 to 18.44 is quite significant! Reducing FID from 100 to 99 is exponentially easier than reducing it from 20 to 19. Consider also that the addition is only one line of code.

**[R4]** **: I would expect the authors to add an ablation study of the comparison using different batchsizes.**

Table 2: The effect of batch-size and Top-$k$ sampling. Top-$k$ sampling's usefulness does not seem to drop off as batch size increases.

| SAGAN (batch-size=64) | Top-$k$ SAGAN (batch-size=64) | SAGAN (batch-size=128) | Top-$k$ SAGAN (batch-size=128) | SAGAN (batch-size=256) | Top-$k$ SAGAN (batch-size=256) |
|---|---|---|---|---|---|
| 21.1 | **19.8** | 19.0 | **17.9** | 18.6 | **17.4** |

[Meta-Review · NeurIPS 2020]

Three reviewers recommend weak accept, while one recommends weak reject. One the one hand, reviewers found the extreme simplicity of the proposed approach appealing and the gains over baseline GANs compelling. On the other hand, many reviewers viewed the novelty as relatively incremental. Additionally, multiple reviewers requested stronger analysis -- e.g. How does the proposed approach affect diversity? -- and comparison with related techniques. Author response largely addressed these questions and concerns. I agree that the simplicity of the proposed method is a strength, rather than a weakness, so long as sufficient analysis is provided. Given the author response, I recommend acceptance. However, I strongly encourage authors to take reviewer feedback seriously and attempt to address all their concerns in camera ready.